# VISUAL REASONING VIA PERCEPTUAL EXTENSION AND IN-CONTEXT LEARNING

## ABSTRACT

The reasoning ability with visual information has recently gained significant attention in the field of large vision-language models (LVLMs). Existing R1-like reasoning LVLMs are usually finetuned from a base LVLM on a large-scale vision-language dataset, incorporating reinforcement learning (RL) with rewards from verifiable answers. However, such reasoning LVLMs usually requires high-quality multimodal long-chain datasets for supervised finetuning in the cold start stage, and time-consuming multiple response sampling in the RL stage. Therefore, we seek to explore an efficient approach to achieve visual reasoning. To do so, we first investigate the interaction between visual and textual tokens in LVLMs, and find that although the post-trained reasoning LVLM improves the cross-modal interaction, but only at deep layers and for long responses, this improvement is negligible for short responses. Based on these observations and insights, we propose to separate the perception and reasoning process, to avoid the LVLM from generating long responses, so that the LVLM maintains cross-modal interaction ability, and do the reasoning by the LLM, which is not required to integrate cross-modal information. To this end, we leverage the existing reasoning large language models (LLMs) with a VLM extension, to synthesize visual and textual information in advance and then perform the reasoning by the LLM, without any finetuning. Furthermore, to make full use of the training samples, we use a matching mechanism to find the relevant reasoning process and incorporate them by in-context learning. We evaluate our method on the common visual reasoning benchmarks. The results show that, without extra training samples, our method achieves performance comparable to the existing post-trained reasoning LVLMs, and outperforms them with in-context learning.

## 1 INTRODUCTION

Reasoning with visual information has become a core capability of large vision-language models (LVLMs) to achieve human-level intelligence (Li et al., 2024; Chen et al., 2023; Bai et al., 2023; Xu et al., 2024; Huang et al., 2025). It enables the model to solve complicated questions that require both visual understanding and multi-step reasoning. Moreover, such reasoning ability also improves the interpretability and trustworthiness of these models. Building on the recent success of reinforcement learning with verifiable rewards in large language model (LLM) reasoning (Guo et al., 2025), existing R1-like reasoning LVLMs are usually finetuned from a base LVLM on a large-scale vision-language dataset (Huang et al., 2025; Chen et al., 2025). However, the training process of reasoning LVLMs usually requires high-quality multimodal long-chain datasets for supervised finetuning in the cold start stage, and time-consuming multi-response sampling in the reinforcement learning stage. This training process requires both a high-quality dataset and massive computational resources. To address these challenges, recent work has investigated more efficient approaches. One line of research leverages existing reasoning LLMs while training only a lightweight projector to align visual and textual tokens (Peng et al., 2025). Although this strategy avoids sampling long-chain responses, it still requires a large number of training samples to train the projector. Given powerful reasoning LLMs such as DeepSeek-R1 (Guo et al., 2025) at hand, we ask: **is it possible to achieve comparable visual reasoning performance without expensive finetuning?**

To this end, we examine the difference between base LVLMs and reasoning LVLMs in terms of vision-text information interaction within attention blocks, aiming to uncover insights for more ef-

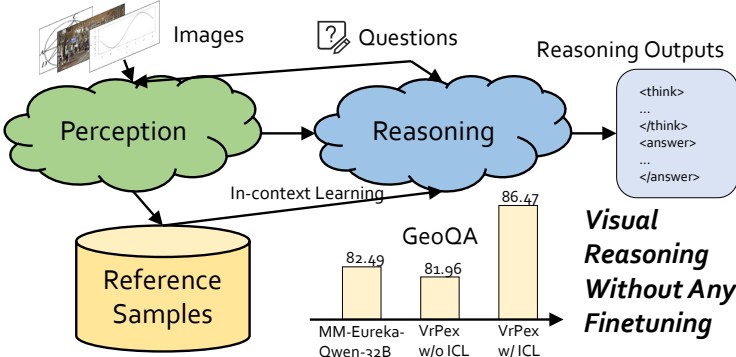

Figure 1: Achieving visual reasoning via a perceptual extension and enhancing it by in-context learning. Our method separates perception and reasoning and does not need any finetuning, while having comparable performance with R1-like reasoning VLMs.

ficient design. Our analysis shows that although reasoning LVLMs does improve the vision-text information interaction when generating long responses. But when generating short responses, this improvement is negligible. Additionally, we find that this improvement occurs only in the deep layers. In the early layers, reasoning LVLMs still exhibit inadequate integration of visual and textual information. We attribute this limitation to the inherent disparity between visual and textual tokens in their semantic abstraction levels: at the early layers, the LVLM prioritizes processing visual tokens, which reduces its capacity to fuse cross-modal information when producing long responses. These observations suggest that LVLMs may need to avoid generating long responses if we want to maintain the same level of cross-modal interaction. Motivated by these findings, we propose to **decouple perception and reasoning**. In this design, the LVLM focuses on perceiving and integrating visual information without the burden of producing long responses, while the LLM performs the reasoning independently. This separation enables the LVLM to more effectively inject visual content into the output, while the reasoning LLM (unconstrained by cross-modal fusion) can handle the complex reasoning process.

Based on these observations and insights, we propose a method to integrate visual information into the reasoning process without requiring any finetuning. Specifically, our approach leverages a powerful reasoning LLM in combination with a VLM extension. The VLM extension first synthesizes visual and textual information, and the reasoning is subsequently carried out by the LLM. Such combination of the reasoning LLM and the VLM extension can be more efficient compared to a single LVLM, since the VLM extension is no longer required to generate long responses where the visual information is difficult to integrate properly. We refer to this framework as **Visual Reasoning via Perceptual Extension (VrPex)**. In addition, we design a matching mechanism to retrieve relevant reasoning trajectories from training samples and incorporate them through in-context learning with the reasoning LLM. This mechanism allows VrPex to exploit available training data while further improving performance on specific benchmarks.

We conduct extensive experiments to evaluate the effectiveness of our proposed method in terms of reasoning performance. The results show that even without access to training samples, VrPex achieves performance comparable to existing reasoning LVLMs. Moreover, when combined with in-context learning, it further outperforms these models. We observe that incorporating relevant training samples into the input context can further enhance performance on targeted tasks, highlighting the extensibility and flexibility of our approach.

## 2 RELATED WORKS

### 2.1 R1-LIKE REASONING LVLMS

Recent advancements of large language models in incentivizing reasoning ability with reinforcement learning (Guo et al., 2025) inspire the researches on R1-like reasoning large vision-language models (Meng et al., 2025; Chen et al., 2025; Huang et al., 2025). These visual reasoning models are usually finetuned with reinforcement learning algorithms (e.g., PPO, DPO, and GRPO) using rewards from verifiable answers. For example, DPO is used in RLHF-V (Yu et al., 2024), LLaVA-Reasoner (Zhang et al., 2024b), and Insight-V (Dong et al., 2025). They construct large-scale preference datasets and directly apply DPO for training. MMPR (Wang et al., 2024) further introduces

a quality loss from a Binary Classifier and a generation loss from supervised finetuning (SFT), in addition to the standard DPO preference loss, thereby strengthening reasoning ability. Meanwhile, GRPO, which has proven effective in DeepSeek-R1, has become a widely adopted RL strategy for reasoning LVLMs. Representative works, including MM-Eureka (Meng et al., 2025), Vt-R1 (Zhou et al., 2025), LLM-R1 (Yingzhe et al., 2025), and R1-V (Chen et al., 2025), apply GRPO to multi-modal reasoning tasks such as mathematical geometry, achieving promising results.

## 2.2 VISUAL IN-CONTEXT LEARNING FOR LVLMS

In-context learning is a paradigm that allows language models to learn new tasks given only a few examples in the form of demonstration (Brown et al., 2020). At inference, in-context learning research has primarily focuses on three aspects: demonstration organization (Zhao et al., 2021; Lu et al., 2021), demonstration selection (Liu et al., 2021; Tanwar et al., 2023; Qin et al., 2023), and demonstration reformatting (Kim et al., 2022; Liu et al., 2023a; Yang et al., 2023). For LVLMs, in-context learning is extended to use visual information as demonstrations (Sun et al., 2023; Liu et al., 2023b). Considering the cross-modal challenge in LVLMs, VICL (Zhou et al., 2024) uses intent-oriented image summary and demonstration composition to address such challenge.

In our paper, we focus on visual reasoning and avoid potential challenge in cross-modal interaction between vision and text by separating perception from the reasoning process. Also, we use utilize the thought process of the similar ones in the training samples as visual demonstrations to improve the reasoning in specific areas, which differs from any of the prior works.

## 3 METHODOLOGY

### 3.1 CROSS-MODAL INFORMATION INTERACTION IN REASONING LVLM

To better understand what makes reasoning LVLMs perform better than base LVLMs and give insights to efficient construction for visual reasoning system, we investigate the interaction between visual and textual information in LVLMs.

Therefore, we look into the attention weights of the base LVLM and the reasoning LVLM trained by RL. Suppose the queries and keys in the attention block at layer $l$ to be $\boldsymbol{Q}^{(l)}, \boldsymbol{K}^{(l)}$. The attention map is

$$\boldsymbol{A}^{(l)} = \text{softmax}\left(\boldsymbol{Q}^{(l)}\boldsymbol{K}^{(l)\top}\right). \tag{1}$$

It can be further decomposed into three parts according to the queries and keys: system prompt tokens ($\boldsymbol{K}_{\text{sys}}^{(l)}$), visual tokens ($\boldsymbol{K}_{\text{vis}}^{(l)}$), and text tokens ($\boldsymbol{Q}_{\text{text}}^{(l)}, \boldsymbol{K}_{\text{text}}^{(l)}$). The attention map can be expressed as a partitioned matrix given the above partition, which is

$$\boldsymbol{A}^{(l)} = \left[\boldsymbol{A}_{\text{text,sys}}^{(l)}, \boldsymbol{A}_{\text{text,vis}}^{(l)}, \boldsymbol{A}_{\text{text,text}}^{(l)}\right]. \tag{2}$$

Specifically, we care about the *proportion of attention weights* that the text tokens attend to the visual tokens (Text-Vision, $\boldsymbol{A}_{\text{text,vis}}^{(l)}$) and the text tokens (Text-Text, $\boldsymbol{A}_{\text{text,text}}^{(l)}$), since they reveal how the information flows before generating the output text.

We use prompts to encourage the model generate long chain reasoning process, then compare the proportion of the attention weights between the base and reasoning models. We use the training samples of GeoQA (Chen et al., 2021) as the inputs. We take Qwen2.5-VL-7B-Instruct (Bai et al., 2025) as the base LVLM, and MM-Eureka-Qwen-7B (Meng et al., 2025) as the reasoning LVLM.

Firstly, we investigate the attention proportion by the response length. For this experiment, we take the average attention proportion across the layers, which is,

$$p_{\text{text,vis}} = \frac{1}{L}\sum_{l=1}^{L}\sum_{i=1}^{N}\boldsymbol{A}_{\text{text,vis},i}^{(l)}, \tag{3}$$

where $N$ is the total number of visual tokens. The Text-Text proportion is similarly calculated.

As we can see in Figure 2a, 2b. We plot the attention proportion against the response length in logarithmic scale. And the dashed lines are the linear fitting of the samples. From the absolute value

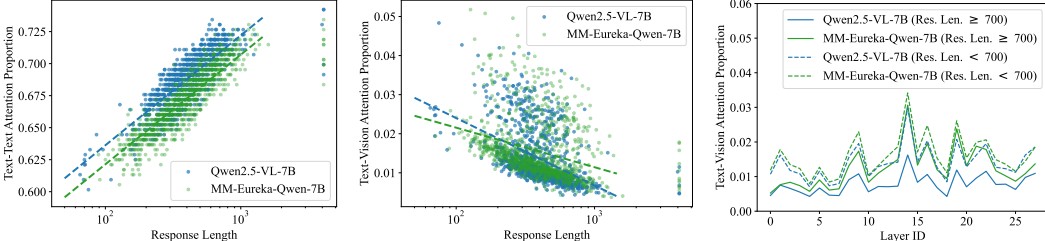

(a) Text-Text Attention Proportion by Response Length  (b) Text-Vision Attention Proportion by Response Length  (c) Text-Vision Attention Proportion by Layer

Figure 2: Investigations on attention proportion of LVLMs. (a)(b) Text-Text and Text-Vision attention proportions are roughly proportional and inverse proportional to the response length. The reasoning LVLM is able to generate longer responses while keeping the attention proportion unchanged. (c) For the reasoning LVLM, the gap between long and other responses diminishes as the layer goes deeper. **However, for short responses, the reasoning LVLM only achieves the same level of cross-modal interaction as the base LVLM**.

of the proportion, Text-Vision takes a much smaller attention proportion than Text-Text, for both base and reasoning models. As the response length increases, the Text-Text takes more attention proportion, and the Text-Vision takes less accordingly. This is reasonable because the longer the response, the more text tokens can be attended to, thus the Text-Text attention proportion rises. More importantly, as for the difference between two models, from Figure 2a, the reasoning model (green) generally generates longer responses, while keeping the same Text-Text attention proportion. From Figure 2b, the Text-Vision attention proportion of the reasoning model diminishes more slowly as the response length increases, compared to the base model (blue). We thus posit that the success of the reasoning LVLMs is related to the increased Text-Vision attention proportion for long responses, which helps the model to integrate the visual information better.

Furthermore, we investigate the attention proportion by layer to find more clues, as shown in Figure 2c. In the figure, we divide the responses into long ($\geq 700$ tokens) and short ($< 700$ tokens) responses and compare the Text-Vision attention proportion by layer. We can see that longer responses usually take a smaller Text-Vision attention proportion for both models, which is consistent with our prior observation. Interestingly, for the reasoning model, the gap between long and short responses reduces as the layer goes deeper, which clearly contrasts to the base model, where the gap remains relatively unchanged. However, the increased attention proportion for long responses at deeper layers for the reasoning model is merely the same as that for short responses. Therefore, we come to another conclusion that although the Text-Vision attention proportion increases for the reasoning LVLM, it mainly occurs at deep layers and for long responses. For short responses, the reasoning LVLM only achieve the same level of cross-modal interaction as the base LVLM. Moreover, considering the difference between visual and text tokens in terms of the abstraction level, such an inadequacy of information integration is likely the fundamental deficiency for the LVLMs to further achieve better performance when generating long responses.

So far, we have found the merits and demerits of the reasoning LVLMs in terms of visual information integration. Although the reasoning LVLM improves the integration of visual information at deeper layers for long responses, it only achieves the same level of integration for short responses. To construct an efficient visual reasoning system, there is no need for the LVLMs to generate long responses if we want to keep the same level of cross-modal interaction. Therefore, we can avoid the VLM from generating long responses, and make it only do the perception to preserve the cross-modal interaction without further post-training. Then to perform the reasoning, we have to incorporate a reasoning LLM to further integrate the information. As such integration is only in the text domain, we do not worry about the cross-modal interaction when generating long responses. It leads us to the separation of the perception and reasoning process.

## 3.2 Perceptual Extension for Reasoning LLM

To keep the interaction between visual and textual information and avoid the deficiency of the LVLMs in generating long responses, we separate the perception from the reasoning process. Then, we use the existing reasoning LLM to generate long responses for reasoning. Specifically, we sepa-

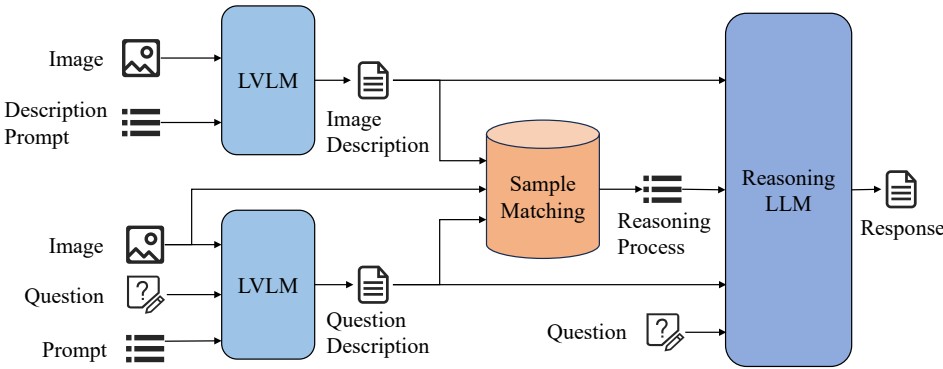

Figure 3: The overview of VrPex with in-context learning. It includes the separation of perception and reasoning, which avoids the inefficient vision-text interaction in the LVLM when generating long responses. It also includes the in-context learning of reasoning process, for performance improvement on specific benchmarks.

rate the perception and the reasoning as follows:

$$o_{\text{desc}} = \text{LVLM}\left(\boldsymbol{x}_{\text{img}}, \boldsymbol{x}_{\text{text}}\right), \quad \text{(Perception)} \tag{4}$$

$$o_{\text{ans}} = \text{LLM}\left(o_{\text{desc}}, \boldsymbol{x}_{\text{text}}\right), \quad \text{(Reasoning)} \tag{5}$$

where we denote the input sample as $(\boldsymbol{x}_{\text{img}}, \boldsymbol{x}_{\text{text}})$, the description text generated by the LVLM as $o_{\text{desc}}$, and the output responses as $o_{\text{ans}}$. Such separation not only avoids the LVLM to generate long responses which the model is probably not good at, but also avoids finetuning on the LVLM which require massive computational resources.

Next, we have to determine what should the description text $o_{\text{desc}}$ be. Naively, the description text can be simply the generated caption of the image. However, such caption is usually general, and shows irrelation to the input question. Thus, the reasoning LLM is rather difficult to extrat relative information from the caption. Therefore, we take a step further to make the LVLM generate targeted description related to the input question. We use carefully designed prompts to make the LVLM perform two generations, one with only the image, the other with both the image and the question, to get the image description and question description. Formally,

$$o_{\text{img\_desc}} = \text{LVLM}\left(\boldsymbol{p}_{\text{img\_desc}}, \boldsymbol{x}_{\text{img}}\right), \tag{6}$$

$$o_{\text{ques\_desc}} = \text{LVLM}\left(\boldsymbol{p}_{\text{ques\_desc}}, \boldsymbol{x}_{\text{img}}, \boldsymbol{x}_{\text{ques}}\right), \tag{7}$$

where $\boldsymbol{p}$ denotes the corresponding prompt.

With the perception results above, the reasoning LLM needs to perform the final reasoning given the related information about the question and the image.

$$o_{\text{ans}} = \text{LLM}\left(\boldsymbol{p}_{\text{reasoning}}, o_{\text{img\_desc}}, o_{\text{ques\_desc}}, \boldsymbol{x}_{\text{ques}}\right). \tag{8}$$

We use existing reasoning LLMs which is trained to generate long responses for the final answers. The visual illustration of VrPex is in Figure 3.

### 3.3 IN-CONTEXT LEARNING OF REASONING PROCESS

So far, we only use the existing LVLMs and reasoning LLMs to construct a system to incorporate visual information in reasoning. In this section, we consider improving the performance further by incorporating the training samples via in-context learning. Therefore, we design a matching mechanism to find the relevant training sample according to the input sample. In the previous section, we use the LVLM to generate the image description and question description for the input sample. We can also use the generated descriptions as keys to find the relevant training samples.

Firstly, we preprocess the training samples for better sample matching and reasoning process output. The raw training samples are usually in the form of image-question-answer triplets. We use GPT-4o (OpenAI, 2025) to generate the structured reasoning process for each training sample, including the image description, question descriptions, and possible reasoning process according to the given answer. The reasoning process does not contain any specific numbers, only contains the general

reasoning steps, in order to exclude the distraction of question-specific information in the training samples.

Then, we leverage a pre-trained image-text encoder to encode the images and texts in the training samples into embeddings. Suppose the training samples to retrieve are represented by $(\boldsymbol{z}_{\text{img}}^{(i)}, \boldsymbol{z}_{\text{img\_desc}}^{(i)}, \boldsymbol{z}_{\text{ques\_desc}}^{(i)}, \boldsymbol{z}_{\text{reasoning}}^{(i)})_i^n$, the encoding can be represented as

$$\boldsymbol{e}_{\text{img}}^{(i)} = \text{Enc}_{\text{img}}\left(\boldsymbol{z}_{\text{img}}^{(i)}\right), \boldsymbol{e}_{\text{img\_desc}}^{(i)} = \text{Enc}_{\text{text}}\left(\boldsymbol{z}_{\text{img\_desc}}^{(i)}\right), \boldsymbol{e}_{\text{ques\_desc}}^{(i)} = \text{Enc}_{\text{text}}\left(\boldsymbol{z}_{\text{ques\_desc}}^{(i)}\right). \tag{9}$$

The embeddings of the training samples can be stored as a database for efficient retrieval at inference. For a specific input sample, we also encode the image and generated descriptions into the embeddings as $(\boldsymbol{e}_{\text{img}}, \boldsymbol{e}_{\text{img\_desc}}, \boldsymbol{e}_{\text{ques\_desc}})$.

We calculate the similarities between the embeddings of the input sample and the training samples in terms of each key. We use the cosine similarity to measure the similarity between the embeddings, which is

$$s_{\text{img}}^{(i)} = \cos\left(\boldsymbol{e}_{\text{img}}, \boldsymbol{e}_{\text{img}}^{(i)}\right), \tag{10}$$

and it is similar for other keys ($s_{\text{img\_desc}}^{(i)}$ and $s_{\text{ques\_desc}}^{(i)}$).

To select the most relevant training samples, we keep the Pareto front of the three similarity measures. Therefore, we obtain a set of the most similar training samples. We then use this set as the demonstrations of in-context learning for the reasoning LLM. The demonstration contains only the reasoning process generated in the sample preprocessing. The number of used demonstrations can be controlled by a hyperparameter $n_d$. Given $n_d$, we select the top-$n_d$ samples according to the total similarity measures. The in-context learning can be represented as

$$\mathcal{D} = \underset{i}{\text{Pareto}}\left((s_{\text{img}}^{(i)}, s_{\text{img\_desc}}^{(i)}, s_{\text{ques\_desc}}^{(i)}), n_d\right), \tag{11}$$

$$\boldsymbol{o}_{\text{ans}} = \text{LLM}\left(\begin{array}{c}\boldsymbol{p}_{\text{reasoning}}, \boldsymbol{o}_{\text{img\_desc}}, \\ \boldsymbol{o}_{\text{ques\_desc}}, \boldsymbol{z}_{\text{reasoning}}^{(i)}, \boldsymbol{x}_{\text{ques}}\end{array}\right), i \in \mathcal{D}. \tag{12}$$

There are cases where the input question and image are not similar to any training samples. This could happen when the type of the question is not covered by the training samples. In such cases, we use a rejection threshold $\lambda$ to filter out the training samples that are not similar enough. If the maximum average similarity is below the threshold, we do not use any training samples as demonstrations.

## 4 EXPERIMENTS

### 4.1 EXPERIMENTAL SETUP

**Benchmarks**. We evaluate our methods on the common visual reasoning benchmarks, including MathVista mini (Lu et al., 2024), MathVerse mini (Zhang et al., 2024a), GeoQA (Chen et al., 2021), and MMK12 (Meng et al., 2025). MathVista and MathVerse are comprehensive math visual reasoning datasets, incorporating various math problems from different areas. GeoQA is more focused on geometry problems, and MMK12 is a math visual reasoning dataset for K-12 students. Most of the questions in the datasets are multiple-choice questions. Mastering such datasets requires the model to be capable of both perception and reasoning.

**Baselines**. We select 4 R1-like reasoning vision-language models as the baselines. They are R1-Onevision-7B, MM-Eureka-Qwen-7B/32B, and Skywork-R1V-38B. R1-Onevision (Yang et al., 2025) and MM-Eureka (Meng et al., 2025) series are both finetuned on curated CoT datasets, then uses reinforcement learning to improve the reasoning ability. Skywork-R1V (Peng et al., 2025) only trains the projector to align the visual and text domains before the reasoning LLM.

**Implementation Details**. It is flexible for our method to use different VLMs and reasoning LLMs. In this paper, we test our method using Qwen2.5-VL-3B/7B-Instruct (Bai et al., 2025) and DeepSeek-R1-Distill-Qwen-7B/14B (Guo et al., 2025) as the VLM and reasoning LLM, respectively. Such combination forms two parameter scales: 10B (3B + 7B) and 21B (7B + 14B).

Table 1: Results on visual reasoning benchmarks. The results are reported in accuracy (%). The best results are highlighted in **bold**. The second bests are underscored.

| Reasoning VLMs | | MathVista | MathVerse | GeoQA | MMK12 |
|---|---|---|---|---|---|
| R1-Onevision-7B Yang et al. (2025) | | 61.3 | 45.94 | 71.75 | 37.50 |
| MM-Eureka-Qwen-7B Meng et al. (2025) | | 73.2 | 50.25 | 80.37 | 60.80 |
| MM-Eureka-Qwen-32B Meng et al. (2025) | | **74.7** | 54.19 | 82.49 | **68.75** |
| Skywork-R1V-38B Peng et al. (2025) | | 70.3 | 46.78 | 73.87 | 48.40 |
| Perception VLM | Reasoning LLM | | | | |
| w/o ICL   Qwen2.5-VL-3B-Instruct | DeepSeek-R1-Distill-Qwen-7B | 71.2 | 48.50 | 73.47 | 51.20 |
| Qwen2.5-VL-7B-Instruct | DeepSeek-R1-Distill-Qwen-14B | 73.4 | 54.03 | 81.96 | 56.65 |
| w/ ICL   Qwen2.5-VL-3B-Instruct | DeepSeek-R1-Distill-Qwen-7B | 73.3 | 51.17 | 81.96 | 60.50 |
| Qwen2.5-VL-7B-Instruct | DeepSeek-R1-Distill-Qwen-14B | 74.1 | **55.99** | **86.47** | 65.25 |

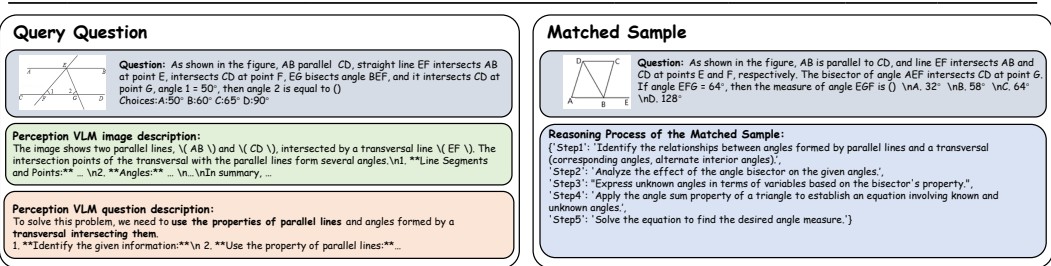

Figure 4: Case study of the proposed method. As shown, in the perception, the image description extracts all of the geometry elements in the image, and the question description gives targeted descriptions of the question. In the sample matching, it finds the sample using the same theorem, with similar reasoning process.

The specific prompts for perception and reasoning is in the supplementary material. For the default setting of in-context learning, we only use the training samples from GeoQA and set the number of demonstrations $n_d$ to 1. On MMK12, we use a subset of the training samples from MMK12 to perform the sample matching. The image-text encoder we use is jina-clip-v2 (Koukounas et al., 2024), which encodes the input to a unified embedding. As stated in the Methodology section, we use GPT-4o to generate the reasoning process for each training sample, and apply a rejection threshold $\lambda$ to ensure the relevance of the training samples when testing on benchmarks in various areas.

## 4.2 BENCHMARK RESULTS

The benchmark results are shown in Table 1. The answers generated by each model is extracted and checked by `math_verify`, which may differ from the results reported in other benchmarks, where they use LLMs to verify the answers. Anyway, it is fair for all of the results in the table. We can see that on most of the benchmarks, VrPex achieves comparable performance with the reasoning LVLMs. On MathVista and MathVerse, VrPex achieves comparable performance with MM-Eureka. On GeoQA, VrPex with ICL outperforms MM-Eureka by a considerable margin. On MMK12, since the samples for matching is sampled from the training set of MMK12, the ICL plays a more significant role. However, MM-Eureka is trained on MMK12, the performance gap is larger compared to other benchmarks. It is notable that VrPex does not require any finetuning, which is easy to construct and deploy. Therefore, our method offers an easier way to achieve visual reasoning with extreme little cost.

We can see that with the in-context learning, VrPex even outperforms some of the reasoning LVLMs in specific benchmarks. Because the samples used for in-context learning are from the training set of GeoQA, which is a geometry reasoning dataset, VrPex has a great performance gain on GeoQA. This shows that the in-context learning can be used to improve the performance on specific benchmarks. This offers more extensibility and flexibility for constructing the visual reasoning solution. In Additional Analysis, we will further investigate the performance gain of in-context learning on other benchmarks in terms of category-wise accuracy.

Furthermore, we explore the flexibility and extensibility of VrPex by using different size of perception VLMs and reasoning LLMs in other scenarios. We perform experiments on perception-intensive

Table 2: Ablation study. We verify the effectiveness of our method by upgrading the method step by step.

| Perception VLM | Reasoning LLM | MathVista | GeoQA |
|---|---|---|---|
| ✓ | ✗ | 55.6 | 50.27 |
| ✗ | ✓ | 48.4 | 68.57 |
| img. desc. only | ✓ | 67.1 | 70.82 |
| ques. desc. only | ✓ | 69.3 | 71.88 |
| img. + ques. desc. | ✓ | 71.2 | 73.47 |

Table 3: Accuracies of different number of demonstrations for ICL. More number of demonstrations does not come with better performance in general.

| Number of Demos | 1 | 2 | 3 | 4 |
|---|---|---|---|---|
| MathVista | 73.4 | 73.2 | 73.3 | 72.9 |
| GeoQA | 81.96 | 81.83 | 82.10 | 81.83 |

reasoning benchmarks like M3CoT Chen et al. (2024) and RealworldQA X.AI (2024). The details are stated in the Appendix C.

## 4.3 CASE STUDY

We illustrate the effectiveness of our method by showing some cases of GeoQA in Figure 4. And leave some detailed cases in the supplementary material.

**Perception VLM** The outputs of the perception VLM are shown in the left part of Figure 4. For the image description, the VLM extracts all of the geometry elements in the image in detail, such as the points, parallel lines, angles, and triangles. For the question description, the VLM gives targeted descriptions of the question, such as the theorems or properties used in the question.

**Sample Matching** The result of the sample matching is shown in the right part of Figure 4. By the three similarity measures used in the matching, it finds the sample using the same theorem, for example, the theorems of the parallel lines and bisect angles. Therefore, the reasoning process of the matched sample can be effectively used as the demonstration for in-context learning.

## 4.4 ABLATION STUDY

To further verify the effectiveness of the proposed method, we perform ablations on the GeoQA dataset. We upgrade the method step by step from only using the VLM to answer the question to using both image and question description. The in-context learning is not applied in the ablations. We use the 10B combination of our method in this experiment. The results are shown in 2.

We first checked the performance of only using the VLM to answer the question and only using the text information (row 1 and 2). From these results, we see that reasoning LLM perform much better on GeoQA than MathVista, so we conclude that MathVista requires more perception than GeoQA. It can further inspire us to use better perception VLM on MathVista to handle perception-intensive benchmarks like MathVista. Then we investigate the performance with only the image description, to show the effectiveness of the question description. The results shows that using question description can improve the performance significantly. Also, we see that using question description only is better than using image description only, but the best performance is achieved by using both of them.

## 4.5 CATEGORY-WISE IMPROVEMENT AFTER ICL

To further investigate the effect of the in-context learning, we look deeper into the category-wise accuracy before and after in-context learning. We take MathVista as an example, showing the results in Table 4 for model scale of 10B. We can conclude that, by using the training set of GeoQA for in-context learning, the accuracy on test samples in geometry-related sources generally increases. It shows that we can specifically improve the capability of the proposed visual reasoning system by using training samples in specific areas with corresponding reasoning process, while not affecting the performance in other areas too much.

Table 4: Category-wise accuracy and relative improvement on Mathvista.

| Sources (Partial) | w/o ICL | w/ ICL | Increment |
|---|---|---|---|
| GeoQA+ (6.2%) | 77.42 | 82.26 | +4.84 |
| Geometry3K (6.2%) | 67.74 | 72.58 | +4.84 |
| UniGeo (6.2%) | 85.48 | 88.71 | +3.23 |
| Super-CLEVER (6.2%) | 56.45 | 54.84 | −1.61 |
| IQTest (3.7%) | 45.95 | 43.24 | −2.70 |
| FigureQA (6.2%) | 64.52 | 61.29 | −3.23 |
| Overall (100%) | 71.2 | 73.3 | +2.1 |

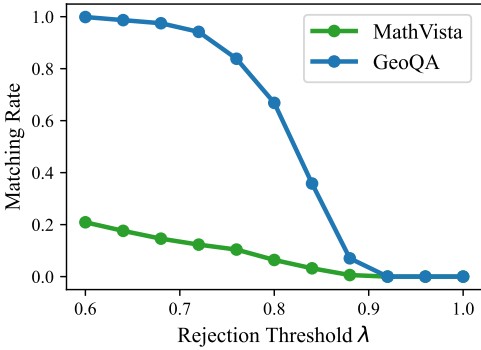 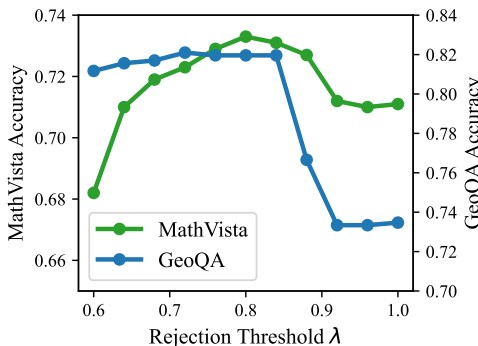

Figure 5: Matching rate is decreasing as the rejection threshold increases.

Figure 6: Benchmark accuracies is affected by the rejection threshold.

### 4.6 HYPERPARAMETER ANALYSIS

In this section, we analysis the effect of two hyperparameters in our proposed method.

**Rejection Threshold for Sample Matching**. The rejection threshold $\lambda$ controls the similarity level that the sample matching accepts the sample as demonstration. We investigate how the rejection threshold affect the accuracy on the benchmarks, the results are shown in Figure 5 and 6.

Matching rate is the proportion of test samples that succeeds matching with at least one sample as the demonstration. We can acknowledge how many test samples actually use the demonstrations with ICL. As we can see from Figure 5, in MathVista, there are many test samples have low similarities (below 0.6) with the matching samples, since we only use the geometry samples for matching and there are other test samples in different areas in MathVista. However, in GeoQA, the similarities are mostly distributed between 0.7 to 0.9. There are much more samples succeed matching with the demonstration as the rejection threshold decreases.

The relation between the rejection threshold and the accuracy on benchmarks are shown in Figure 6. The accuracies generally rises and then drops as the rejection threshold increases. We can choose the best threshold at around 0.8.

**Number of Demonstrations for ICL**. For the in-context learning, we can take multiple samples from the pareto front of three similarity measures as the demonstrations. We perform experiments with different number of demonstrations $n_d$ on MathVista and GeoQA. The results are shown in Table 3. We can see from the results that the number of demonstrations does not affect the performance too much. It can be explained by the similar reasoning processes of the retrieved samples.

## 5 CONCLUSIONS

In this paper, we explore an efficient approach to achieve visual reasoning without any finetuning. We first investigate the differences between base VLMs and existing R1-like reasoning VLMs in terms of the interaction between visual and textual tokens. Our study reveals that while the reasoning LVLM exhibits stronger cross-modal interaction than the base model, this improvement mainly occurs in deeper layers and when generating long responses. For short responses, the reasoning LVLM achieves a level of interaction comparable to the base VLM. Motivated by these findings, to avoid finetuning and deficiency of the base VLM on generating long responses, we separate the perception and reasoning process in visual reasoning. We thus propose Visual Reasoning via Perceptual Extension (VrPex), which uses a VLM to generate both image and question description, followed by a reasoning LLM to perform the reasoning. Furthermore, to make full use of the training samples, we incorporate in-context learning in the reasoning LLM by matching relevant reasoning processes from training samples. The experiments show that VrPex can achieve performance comparable to R1-like reasoning VLMs, and with in-context learning it outperforms them, while avoiding the costly finetuning process.

**Limitations**: Although the proposed approach can achieve visual reasoning without any finetuning, the perception capability relies on the perception VLM, and the reasoning capability relies on the reasoning LLM. To get better perception and reasoning capability, it needs to use larger models and more test-time computation. Although the ICL can offer some flexibility and further improvement for the system, it requires to preprocess the training samples first.

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

## A    PROMPTS FOR PERCEPTION AND REASONING

Here we provide the prompt template for the perception VLM and the reasoning LLM.

Image description prompt $p_{\text{img\_desc}}$:

*You are an image descriptor. When the user gives you an image, you need to describe the image in precise description, revealing the relations of all the elements in the image.*

Question description prompt $p_{\text{ques\_desc}}$:

*You are a question solving assistant. When the user gives you an image, along with a question, you need to output an effective prompt for the next large language model to get the correct answer. The prompt must describe the image in precise description, revealing the relations of all the elements in the image, and point out the possible concepts relating to the question.*

Reasoning prompt template $p_{\text{reasoning}}$:

*Here is the image description: $\{o_{img\_desc}\}$ and question description: $\{o_{ques\_desc}\}$ Think as carefully and methodically about the problem as you need to. Referring to the following demonstrations and hints: $\{z_{reasoning}\}$ Give a true answer to the following question: $\{x_{\text{ques}}\}$.*

## B    DETAILED CASE STUDIES

Here we show some cases of VrPex with ICL in detail. The samples are from GeoQA, and the model combination is the 10B one. The results are shown in Figure 7, Figure 8, and Figure 9. We can frequently observe the reasoning model corrects the answer given by the perception VLM, and outputs the correct answer. Even though the image descriptions are not always precise, it can help the matching process to find the best matching sample. And thanks to the three similarity measures, we can easily find the most related samples in the training set with the same reasoning process.

As we can see, the perception VLM and the reasoning model work together to get the final answer. In VrPex, the reasoning model rarely get confused by no sufficient information provided, if there is enough information in the question or the descriptions. However, for those questions that require stronger perception capability, we can use larger VLM, or train better perception VLM. On the other hand, to enhance the reasoning capability, we can use or train better reasoning LLM, both ways avoid training reasoning VLMs.

## C    PERFORMANCE ON PERCEPTION-INTENSIVE REASONING BENCHMARKS

In Section 4.2, we test VrPex on several visual reasoning benchmarks. However, these benchmarks are relatively biased to text reasoning, and do not require strong perception capability. In this section, we test VrPex on some perception-intensive reasoning benchmarks, including M3CoT Chen et al. (2024), RealworldQA X.AI (2024). These benchmarks contains reasoning questions in real world scenarios, requiring the model to frequently refer to the image to get the correct answer. To deal with these benchmarks, we use a larger perception VLM in VrPex, which is Qwen2.5-VL-32B-Instruct, and use DeepSeek-R1-Distill-Qwen-7B as the reasoning LLM. The results are shown in Table 5. As We can see, VrPex achieves comparable visual reasoning capability in visual-intensive scenarios, which shows the extensibility of VrPex, that we can easily improve the perception capability by using larger and better VLMs.

Table 5: Results on perception-intensive reasoning benchmarks. The results are reported in accuracy (%). VrPex also achieves competitive performance on these benchmarks with a larger VLM.

| Reasoning VLMs | | M3CoT | RealworldQA |
|---|---|---|---|
| R1-Onevision-7B Yang et al. (2025) | | 46.89 | 37.38 |
| MM-Eureka-Qwen-7B Meng et al. (2025) | | 36.97 | 44.05 |
| MM-Eureka-Qwen-32B Meng et al. (2025) | | 38.57 | 46.14 |
| Skywork-R1V-38B Peng et al. (2025) | | 34.21 | 35.03 |
| Perception VLM | Reasoning LLM | | |
| Qwen2.5-VL-32B-Instruct | DeepSeek-R1-Distill-Qwen-7B | 49.87 | 47.58 |

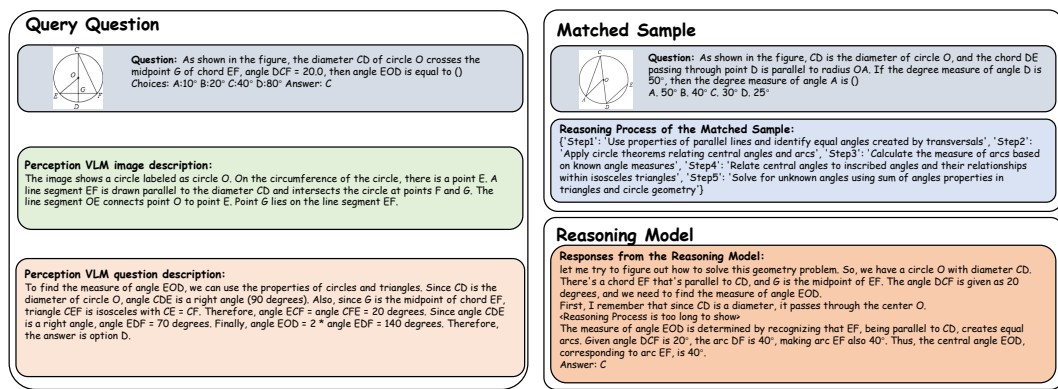

Figure 7: Detailed case 1. We also find that the reasoning model corrects the answers given by the perception VLM. Also, the matched sample is highly related to the input one, with the element of inscribed angles.

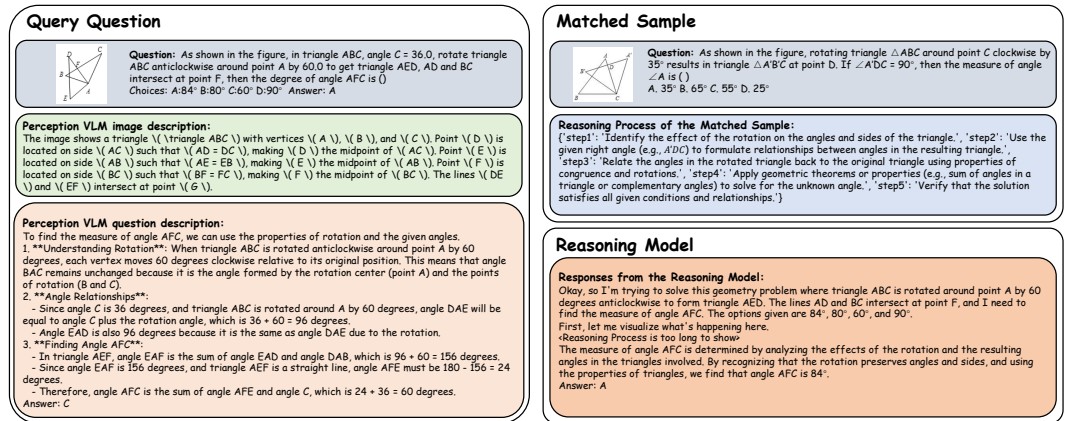

Figure 8: Detailed case 2. We find that the reasoning model corrects the answers given by the perception VLM. Also, the matched sample is highly related to the input one, with the element of rotating triangles.

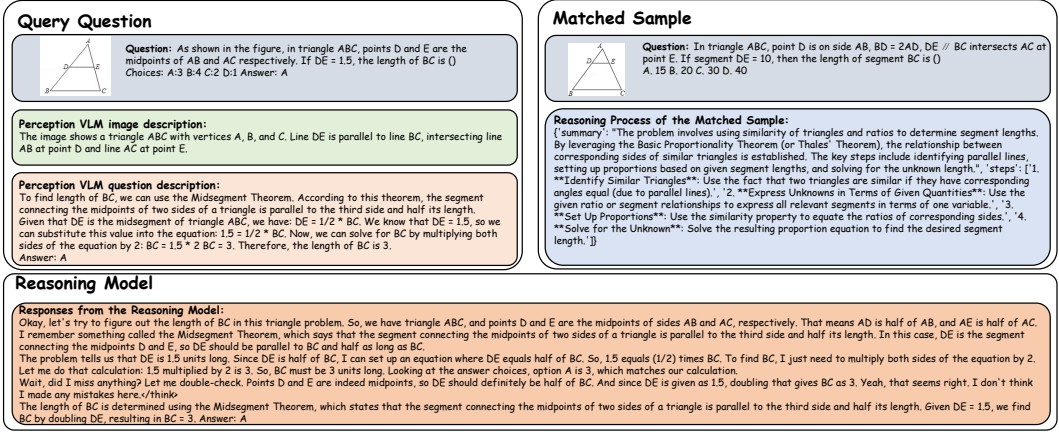

Figure 9: Detailed case 3. The matched sample is highly related to the input one, with the element of similar triangles.

