# OpenReview forum: "Visual Reasoning via Perceptual Extension and In-Context Learning"
_ICLR.cc/2026/Conference — Submitted to ICLR 2026_

### Official Review · Reviewer_ZQ6J · 2025-10-17

**Soundness:** 3
**Presentation:** 2
**Contribution:** 2
**Rating:** 4
**Confidence:** 3

**Summary:**

The authors proposed a training-free multimodal reasoning system. This system uses a multimodal large language model for perception, then passes the perceived information to a large language model with strong reasoning capabilities for inference. To further enhance reasoning capabilities, the authors establish a sample library and retrieval samples from it for incontext learning.

**Strengths:**

1. A simple and effective method that improves reasoning capabilities in multimodal domains.

2. The proposed method improves the performance on a series of mathematics-related multimodal datasets.

**Weaknesses:**

1. The author uses too many subordinate clauses, which makes the article difficult to read.

2. There is a lack of clear connection between the phenomena observed by the authors and the proposed method. (detailed in questions)

3. The proposed ICL method is not novel, similar methods are widely applied in LLMs [a]

[a] Ling Yang, Zhaochen Yu, Tianjun Zhang, et.al. Buffer of Thoughts: Thought-Augmented Reasoning with Large Language Models

[b] Siru Ouyang, Jun Yan, I-Hung Hsu, et. al. ReasoningBank: Scaling Agent Self-Evolving with Reasoning Memory.

**Questions:**

1. In lines 193-194, the author argues, ``however, that the increased attention proportion for long responses at deeper layers for a signaling model is merely the same as that for short responses.'' This argument lacks support from specific experimental observations.

2. The authors argue that the base MLLM and the reasoning MLLM are identical for short responses, and therefore can be used for perception. However, as shown in Fig-2(b), the reasoning MLLM still acquires visual information for long responses. These perceptions are absent when using the LLM for reasoning. The authors' experiments fail to demonstrate that these interactions are unimportant, and the proposed method fails to compensate for them.

3. Since LLM has the potential risk of memorizing answers, I suggest that the authors add a set of experiments in which only the original questions (excluding images) are input into the Reasoning LLM to further demonstrate the effectiveness of the method.

---

### Official Review · Reviewer_3FnV · 2025-10-24

**Soundness:** 2
**Presentation:** 2
**Contribution:** 2
**Rating:** 2
**Confidence:** 4

**Summary:**

This paper presents a method to integrate pretrained LVLM with reasoning LLMs to perform reasoning on vision-language tasks. The approach differ from reasoning LVLM in that it doesn't require additional CoT training data for RL and can leverage SoTA LVLM and reasoning LLMs out of the box. Furthermore, it synthesize and sample reasoning examples for in context learning. Strong performance on visual reasoning benchmarks demonstrates the effectiveness of this training free method to improve visual reasoning.

**Strengths:**

* The integration of ICL with reasoning model improves performance on reasoning benchmarks, and ablations demonstrate the effectiveness of the rejection sampling used for selecting ICL examples.

* The method requires not training and benefit from new reasoning LLM and LVLMs to further improve the performance.

**Weaknesses:**

* There is no justification on why to use proportion of attention weight to measure visual information integration. The paper lacks a clear definition and evidence on what visual information integration means, how that connects to attention weights, and how it affects downstream performance, as having higher attention weights doesn't immediately imply that the visual performance is better.

* The paper argues that the success of reasoning LVLM is related to their higher vision-language attention proportion compared to base models in long responses. At the same time, the paper points out that the response from reasoning models are generally longer, and that shorter response has higher attention proportion. Based on this argument, the base model should be superior than the reasoning models in short responses. Experiments on lengths vs performance would be needed to support these claims.

* The method lacks novelty, as it is similar to prior works that uses LVLM as captioner and LLM as the reasoner$[$Chen et al., 2023$]$.

### References
- Chen, L., Li, B., Shen, S., Yang, J., Li, C., Keutzer, K., Darrell, T., & Liu, Z. (2023). *Large language models are visual reasoning coordinators*. Advances in Neural Information Processing Systems, 36, 70115–70140.

**Questions:**

N/A

---

### Official Review · Reviewer_aHVm · 2025-10-31

**Soundness:** 2
**Presentation:** 2
**Contribution:** 1
**Rating:** 2
**Confidence:** 5

**Summary:**

This paper focuses on visual reasoning with large vision-language models more efficient. The authors first studied how base LVLMs and reasoning LVLMs interact with visual and text info, finding that reasoning LVLMs only get better at cross-modal interaction in deep layers and for long responses. So they propose VrPex, a method that splits perception and reasoning: the VLM handles describing images and questions, and a separate reasoning LLM does the actual reasoning. They also added in-context learning by matching relevant training samples’ reasoning processes to boost performance. The results show VrPex matches existing reasoning LVLMs without any finetuning, and even outperforms them on benchmarks like GeoQA when using in-context learning.

**Strengths:**

1. One big strength is how the authors identified key limitations of current reasoning LVLMs through careful analysis of attention weight: their findings about cross-modal interaction only working for long responses in deep layers really drive the method’s design.

2. The proposed VrPex is super efficient. It skips expensive finetuning entirely, which saves a ton of computational resources, yet still delivers strong results.

3. The paper is well-written and easy to follow.

**Weaknesses:**

1. I appreciate the authors' effort in identifying the problems in VLMs using attention. But the same idea has been verified in many papers, which combines a VLM and an LLM to boost reasoning, like in R1-Onevision, they use this method to collect high quality reasoning data and finally transfer this capability into VLM. The author re-invent this through a very clever way but did not dive into it, just resulting in the combining strategy, making the paper contribution poor.

2. Regarding ICL, in-context learning has long been verified effective in various domains. The author mainly contributes a pipeline to collect high-quality ICL examples. But what makes ICL really impressive is that it can help the model to generalize. However, the author maintain the scope within a limited set, making it not that generalize.

3. The method combines two models in a training free manner to boost reasoning capabilities, the authors should include more pairs of perception models and reasoning models to demonstrate the generalization of the proposed framework.

**Questions:**

As stated in the weakness part.

---

### Official Review · Reviewer_trBg · 2025-11-05

**Soundness:** 3
**Presentation:** 3
**Contribution:** 2
**Rating:** 2
**Confidence:** 4

**Summary:**

This paper proposes VrPex (Visual Reasoning via Perceptual Extension), a training-free approach to visual reasoning that separates perception and reasoning into two stages: a VLM generates image and question descriptions, followed by a reasoning LLM that processes these descriptions to produce answers. The motivation comes from analyzing attention patterns in reasoning LVLMs, which reveals that cross-modal interaction improves primarily in deeper layers for long responses but remains comparable to base models for short responses. The authors further enhance performance through in-context learning by retrieving similar reasoning trajectories from training data. Experiments on mathematical reasoning benchmarks (MathVista, GeoQA, MMMU, MMK12) show that VrPex achieves performance comparable to finetuned reasoning LVLMs like MM-Eureka.

**Strengths:**

The method requires **no finetuning yet achieves results comparable to extensively trained reasoning LVLMs**. This is practically valuable and demonstrates that modular approaches can be effective alternatives to end-to-end training. Section 3.1's investigation into how reasoning LVLMs handle visual-textual interaction across layers and response lengths provides **useful empirical findings to drive work in the field**. The observation that improved cross-modal fusion occurs primarily in deep layers for long responses is interesting and motivates the architectural choice. The modular design allows **easy swapping of perception and reasoning components** (as shown in Table 5 with different model sizes), and the **in-context learning mechanism provides task-specific improvements without retraining**. The paper includes proper ablations (Table 2, 4), hyperparameter analysis (Figures 5-6, Table 3), and category-wise performance breakdowns (Table 4).

**Weaknesses:**

1. **Insufficient acknowledgment and differentiation from prior work.** The core idea of decoupling visual perception from linguistic reasoning by using text descriptions as an intermediate representation is not novel. Several prior works have explored this exact paradigm: LENS (2023) uses frozen VLMs to generate textual descriptions (tags, attributes, captions) and feeds them to frozen LLMs for reasoning without any training. PICa converts images to captions/tags for GPT-3 processing in visual question answering. PNP-VQA uses image captions to aid LLMs in zero-shot VQA; and ViperGPT and variants (Toolformer, Chameleon, HuggingGPT, and Modular Visual Question Answering via Code Generation) leverage vision modules feeding into separate language models for visual reasoning tasks. The paper cites none of these directly relevant works and does not clearly articulate what distinguishes VrPex. The attention analysis provides motivation but doesn't fundamentally change the approach. Without explicit comparison to these or similar baselines or discussion of what VrPex contributes beyond prior modular vision-language systems, the novelty claim is undermined.

2. **Overly broad problem framing undermines contributions.** The title and introduction frame this as solving "visual reasoning," but the experiments evaluate only on multimodal mathematical reasoning benchmarks (GeoQA, MathVista, MathVerse, MMK12). In the broader vision community, visual reasoning encompasses diverse tasks like visual question answering, image captioning, classification, retrieval, spatial reasoning, object detection, and commonsense reasoning. The paper's narrow focus on geometry and math problems doesn't support the broad claims.

3. **Evidence suggests visual information is not critical for the evaluated tasks.** The paper's own results reveal that the benchmarks used don't strongly require visual reasoning. In Table 2, using only the reasoning LLM (without any visual information) achieves 68.57% on GeoQA, compared to 73.47% with full VrPex: only a 5% difference. This suggests GeoQA is heavily language-biased and can be largely solved from text alone. Figure 4's examples reinforce this: both geometry problems provide textual descriptions sufficient to solve them without images (e.g., "AB is parallel to CD" is stated explicitly). If the core motivation is improving cross-modal interaction for visual reasoning, demonstrating this on benchmarks where visual information is actually necessary would be far more convincing.


4. **Limited evaluation scope.** If the paper wishes to claim this as a contribution to general visual reasoning, more evaluation is required. Otherwise, the claims should be altered and simplified. Beyond mathematical reasoning, the paper should evaluate on vision-grounded benchmarks like Winoground, BLINK, or visual spatial reasoning tasks where visual information is actually necessary.

**Questions:**

The weaknesses section incorporates many open questions I had about the work. The feedback there is constructive in the sense that I both lay out the concern and a very clear sense of what needs to be done to address the issue if possible.

---

### Meta-Review · Area_Chair_utc2 · 2025-12-30

**Summary:**

All four reviewers independently identified that decoupling VLM perception from LLM reasoning is well-established. Prior works (e.g., PNP-VQA, ViperGPT) explore this exact paradigm. The paper cites none of these directly relevant works. Besides, the paper claims to address visual reasoning broadly, but evaluates only on mathematical reasoning benchmarks. The authors' own Table 2 suggests the benchmarks are language-biased. The connection between the attention analysis and the method is unclear.

**Reviewer Concerns:**

The authors' decision not to submit a rebuttal leaves all concerns standing.

**Reviewer Scores:**

While the paper presents a simple, effective, and practical approach with solid experimental results, the consensus among reviewers regarding limited novelty is well-founded.  The core idea of decoupling VLM perception from LLM reasoning is well-established in the literature, and the attention analysis does not constitute sufficient methodological innovation to overcome this. This paper received a consensus reject decision from all four reviewers.

---

### Decision · Program_Chairs · 2026-01-26

Reject